# Organoids as Innovative Models for Bone and Joint Diseases

**DOI:** 10.3390/cells12121590

**Published:** 2023-06-08

**Authors:** Jie Huang, Lingqiang Zhang, Aiping Lu, Chao Liang

**Affiliations:** 1Department of Biology, School of Life Sciences, Southern University of Science and Technology, Shenzhen 518055, China; 2Institute of Integrated Bioinfomedicine and Translational Science (IBTS), School of Chinese Medicine, Hong Kong Baptist University, Hong Kong SAR, China; 3State Key Laboratory of Proteomics, National Center for Protein Sciences (Beijing), Beijing Institute of Lifeomics, Beijing 102206, China; 4Institute of Arthritis Research in Integrative Medicine, Shanghai Academy of Traditional Chinese Medicine, Shanghai 200052, China; 5Guangdong-Hong Kong-Macau Joint Lab on Chinese Medicine and Immune Disease Research, Guangzhou 510120, China

**Keywords:** stem cells, bone organoids, joint/bone disease, drug screening, precision medicine

## Abstract

Bone is one of the key components of the musculoskeletal system. Bone and joint disease are the fourth most widespread disease, in addition to cardiovascular disease, cancer, and diabetes, which seriously affect people’s quality of life. Bone organoids seem to be a great model by which to promote the research method, which further could improve the treatment of bone and joint disease in the future. Here, we introduce the various bone and joint diseases and their biology, and the conditions of organoid culture, comparing the in vitro models among 2D, 3D, and organoids. We summarize the differing potential methods for culturing bone-related organoids from pluripotent stem cells, adult stem cells, or progenitor cells, and discuss the current and promising bone disease organoids for drug screening and precision medicine. Lastly, we discuss the challenges and difficulties encountered in the application of bone organoids and look to the future in order to present potential methods via which bone organoids might advance organoid construction and application.

## 1. Introduction

### 1.1. Bone and Joint Biology

Bone, one of the key components of the musculoskeletal system, can support locomotion and provide organ protection, and some bones also have functions in hematopoiesis and help to maintain mineral homeostasis [1]. Adult bone is composed of cortical bone and trabecular bone: the former, with blood vessels and osteocytes, is a compact bone covering the outer bone surface; the latter, containing bone marrow or fat, constitutes the inner bone (Figure 1) [2,3]. The outer cortical bone is sound mechanically and provides protection and support, whereas the inner trabecular bone is active in metabolism and is related to hematopoiesis within the bone marrow niche. It is essential to understand the primary morphology and components of bone since each element of bone plays different functions in bone remodeling, growth, and development. Microscopically, four characteristic cell categories can be found (osteoclasts, osteoblasts, osteocytes, and osteoprogenitor cells) [4]. Osteoclasts are generally multinuclear giant cells originating from hematopoietic stem cells that migrate from the bone marrow to the bone surface, and they are responsible for bone degradation during the absorption phase of bone remodeling via direct chemical and enzymatic attack. Osteoblasts from osteoprogenitor cells, which migrate to the remodeling location, synthesize and secrete bone matrix (osteoid) and regulate the deposition of minerals during the formation phase of skeletal remodeling. Osteocytes, which derive from osteoblasts and account for 90% of all bone cells, can detect mechanical loading. Osteoprogenitor cells have the ability to differentiate into osteoblasts and are responsible for the formation of bone remodeling compartments within a specialized microenvironment [4].

The indirect connection between two bones is called the joint, which is generally composed of the joint surface, the joint capsule, and the joint cavity [5]. The joint surface is covered with cartilage, which is composed of strong and elastic connective tissues to reduce friction, vibrations, and shocks during exercise. The joint capsule includes two layers: the outer layer, which is a fibrous layer that enhances joint stability; and the inner layer, which is a synovium layer secreting synovial fluid in order to reduce damage during movement and sports. The joint cavity is a narrow space around the joint cartilage and capsule, which normally contains only a small amount of synovial fluid (Figure 1). Various cells in joint anatomy play irreplaceable roles in physiological activity [5]. Articular cartilage just contains the chondrocytes, which function as the bearing surface of the joint [6]. Around the joint synovium, fibroblast-like synoviocytes (FLS), macrophage-like synoviocytes (MLS), and vascular endothelial cells (VEC) can be observed [7]. The strength and integrity of bone is subject to a continuous and dynamic remodeling process via bone resorption and formation in order to maintain a healthy bone mass and growth in healthy physiology. The imbalance between osteoblasts/osteoclasts in bone remodeling or between fibroblast/macrophage synoviocytes in articular joints induces bone/joint-related diseases such as osteoporosis or osteoarthritis. There are many common bone and joint disorders, such as fracture, osteoporosis, osteomyelitis, osteosarcoma, osteoarthritis, and rheumatoid arthritis, that affect people’s quality life seriously (Table 1). Therefore, understanding bone construction contributes to the determination of bone/joint disease etiology and treatment strategies.

### 1.2. Comparison of Research Models In Vitro

The World Health Organization (WHO) shows that musculoskeletal diseases are the leading contributor to disability, and approximately 1.71 billion people have musculoskeletal diseases worldwide. As a crucial member of musculoskeletal diseases, BJD has become the focus of attention. The WHO lists bone and joint disease (BJD) as the fourth most widespread disease, in addition to cardiovascular disease, cancer, and diabetes. The WHO estimates that by 2025, the global population with arthritis will exceed 800 million, second only to cardiovascular-related diseases (accessed on 3 January 2023, https://www.who.int). Therefore, it is necessary to improve bone and joint disease treatments both in terms of efficacy and safety.

Through the unremitting exploration of researchers in in vitro and animal models, various medicines and surgical methods have been shown to relieve the symptoms of BJD, such as pain, swelling, and inflammatory response. However, it is challenging to mimic bone in vitro because the bone always stays in dynamic equilibrium due to the balance of osteoclasts and osteoblasts. Simple models such as 2D cultures are great models for studying cell functions in vitro, but it is difficult to truly simulate cell functions and signaling pathways in vivo without creating cell–cell and cell–matrix interactions. The phenotype of primary cells from purified populations could be lost when cultured in a 2D system. However, 3D cell aggregate cultures of mesenchymal stem cells (MSCs) or tumor cells have exhibited improved function [8,9], despite the fact that they lack the associated tissues that are present in vivo. Cell spheroids culture is another form of 3D culture system that lacks stem or progenitor cell populations, resulting in the inability to differentiate and self-renew in a cell spheroids culture system. Additionally, animal models, the most popular experimental model, have a close similarity to in vivo human physiology, and can thus help to test the efficacy and safety of new treatment strategies; yet, many drugs still fail during clinical trials due to species diversity [10,11].

Moreover, a major obstacle to developing effective treatments for some BJDs, such as RA, is the lack of an accurate form of experimental platform via which to identify new treatments and test their efficacy; this is especially the case for non-responders and the need to personalize their treatment. As an emerging alternative, organoid culture has been a popular and effective model by which to study development, biology, drug screening, and disease modeling in vitro [12], which is closer to in vivo physiology. This is because organoids not only include a large number of biological parameters, such as the spatial organization of heterogeneous tissue-specific cells, cell-matrix interactions, and cell-cell interactions, and but also include special physiological functions generated by tissue-specific cells. The gap between the existing model systems and those in vivo can be greatly minimized by organoids providing a stable cultivated system, which means that organoids are more representative of physiological activity in the body. Here, we will summarize the organoid culture, bone organoid culture from stem cells, and the applications of bone disease organoid models.

## 2. Organoid

Organoids are in vitro, living, stem cell-derived, organ-specific three-dimensional (3D) clusters of self-organized cells that are capable of maintaining the function of organs of origin and aspects of molecular and cellular heterogeneity. The stem cells can be obtained from human cells cultured in vitro, or in vivo, in chimeric animals [13]. In 2009, the first organoid was established; this was an intestinal organoid from the Lgr5+ stem cell recapitulates structures of distinct intestinal crypts and villus domains containing various differentiated cells of intestine in vivo [14]. Since then, numerous organoids have been successfully cultured, including from the brain, retina, kidney, liver, lung, and prostate [15,16,17,18,19,20].

### 2.1. Cell Sources for Organoid Culture

Organoids can be derived from pluripotent stem cells (PSCs) and various adult stem cells (ASCs) [21,22]. PSCs, both embryonic stem cells (ESCs) and induced pluripotent stem cells (iPSCs), can differentiate into any cell type in the body under special conditions. It should be noticed that iPSCs are somatic cells that are restored to a pluripotent state by cellular reprogramming [23]. Organoids from PSCs are acquired by simulating the sequential signaling interactions that drive development in vivo, while ASC-derived organoids are obtained by providing signal stimulates that act in the corresponding adult tissue. PSCs have a broader potency, allowing the coordinated production of cells from multiple germ layers [21]. In contrast, the potency of ASCs is limited and is relevant to the tissue of origin, so guiding the formation of ASC organoids typically does not require extensive sequential switching through different lineage-specific culture conditions. In the two sources of stem cells mentioned above, ASC-based organoids are more relevant to the homeostasis conditions, developmental and regenerative processes of the corresponding organs, whose microstructures are more like that of adult tissues [24]. Up to now, both PSCs and ASCs can be cultured to form various organoids, such as those from the mammary gland, liver, kidney, pancreas, stomach, intestine, lung, and endometrium. ASCs have been proven to culture bladder and prostate organoids [25,26]. In addition, PSCs, not ASCs, have been shown to form brain, optic cup, thyroid, blood vessels organoids, and so on [27,28,29,30].

### 2.2. Organoid Culture Techniques

Different methods have been discovered to provide a cultivated environment for various tissue-specific organoids, including 3D scaffolds and biochemical factors. There are three different scaffolds that can be used to support organoid development. The most common way is to use a solid extracellular matrix (ECM). Matrigel, which comprises laminin, IV collagen, and nest protein, is the substitute of natural ECM purified from Engelbreth–Holm–Swarm mouse sarcoma and is the most widely used matrix for 3D cell culture and organoid cultivation [31]. Intestinal, cerebral, gastric, and mammary gland organoids are only some of the examples that have been successfully generated using Matrigel [14,32,33,34]. Collagen type I, the main component of ECM, is also used to contribute to the derivation of mammary gland and intestinal organoids [35,36]. These natural matrices contain ECM components and growth factors that can promote cell growth and differentiation efficiently, which is the biggest advantage compared with other matrices. However, these matrices are difficult to control in a cultivated environment and during repeat experiments due to the variability and complexity of the components in their natural state. Surprisingly, synthetic hydrogels, which have been added into organoid culture (cerebral and intestinal) to replace natural matrices, solve these problems, [37,38]; however, these hydrogels need to be custom tailored to meet specific requirements in different organoids owing to their low bioactivity.

The air–liquid interface technique has also been applied for organoid culture. The organoids of the kidney, intestine, gastric system, liver, pancreas, lung, and bladder have been generated by applying this technique. In this method, cells are cultured in the form of a pellet on a thin microporous membrane, with cell culture medium only on the base side of the membrane. The cell pellet then self-organizes into a multilayer structure, similar to the microstructure of a natural organ. [25,39,40,41,42,43,44]

Another method employed involves the culture of 3D cell aggregates in suspension, and has been adopted for the formation of cerebellar, optic cup, cerebral, and hippocampal organoids [45,46,47,48]. The growth and development of cell aggregates take place in a solid scaffold-free manner in the suspension culture, but in some cases, to promote the formation of polarized epithelial structures, low concentrations of Matrigel are added.

### 2.3. Organoid Culture Techniques

Because stem cells do not have the ability to form physiological and elaborately relevant tissue under all conditions, stimulation factors, in addition to the scaffold, are also an indispensable part of the 3D organoid culture system, and the simulation of these factors in a culture system helps stem cells to develop, proliferate and differentiate into specific cell types. As shown in Figure 2, PSC-derived organoids first involve three germ-layer specifications (endoderm, mesoderm, or ectoderm) that use different factors. Adding activin A, activin A and bone morphogenetic protein4 (BMP4), Wnt and BMP4 to PSC can form an endoderm, mesoderm, and ectoderm, respectively (Table 1 and Table 2). Then, the three germ layers are induced to differentiate and mature by culturing them with specific growth and signaling factors in order to obtain the specific cell types that form the desired organ. In the cultivation of most PSC-derived organoids, the Wnt and fibroblast growth factor (FGF) signaling pathways are activated via the addition of Wnt-3A and FGF, respectively, whereas the BMP pathway is inhibited by noggin (NOG) [49]. For example, the common method used to culture human gastric organoids from PSCs involves the application of a special medium containing the combination of biochemical factors, such as Wnt-3A, FGF4, NOG, epidermal growth factor (EGF), retinoic acid (RA), CHIR99021, and Y-27632, at different stages in a basement matrix [33]. However, in the PSC-derived human liver organoid culture system, Wnt3A is not in the medium [50]. In bone organoids, growth factors such as CHIR99021 and retinoic acid also play a key role in the differentiation of iPSCs and ESCs into mesodermal lineage [51].

Unlike PSC-derived organoids, in ASC-derived organoid cultures, the tissue-specific stem cell population is required to isolate, which is embedded into an ECM with defined, various growth factors to allow proliferation. In the cultivation of most ASC-derived organoids, Wnt, FGF, and EGF signaling pathways are activated via the addition of Wnt-3A, FGF, and EGF, respectively, whereas BMP and transforming growth factor (TGF) pathways are inhibited by the supplement of NOG and A83-01, respectively, [49] (Table 2 and Table 3). Gastric and lung organoids are examples. ASC-derived human gastric organoids are cultured in medium containing EGF, NOG, R-spondin 1, Wnt-3A, FGF-10, gastrin, A83-01, CHIR99021, prostaglandin E2 (PGE2), Y-27632, SB202190, and insulin-like growth factor (IGF), but in the ASC-derived human lung organoid culture system, EGF is not in the medium to induce the bind with EGF receptors.

To sum up, there are three crucial points in the process of organoid formation, including bone organoids. At the beginning, to establish the correct regional identity during stem cell differentiation, some commercial signalling inhibitors or activators are supplied in order to inhibit or activate the key signalling pathways that regulate developmental patterning. Moreover, the media formulations should allow the appropriate terminal differentiation of the required cell types within the organoid to be developed, usually following the established approach used for the 2D culture or inspired by the murine developmental process. Finally, the way that cultures are grown allows them to expand in a 3D space, which is obtained either by aggregating cells into 3D structures or by embedding cells into a 3D matrix.

Although the process of bone organoid formation is similar to that of other organoid formations, the process of bone organoid formation has higher requirements in many aspects, such as a special medium content, matrix, and mechanical stimulation. For example, Akiva A et al. [52] successfully employed an osteogenic medium containing ascorbic-acid-2-phosphate, dexamethasone, and β-glycerophosphate to culture woven bone organoids.

**Table 2 cells-12-01590-t002:** Summary of organoids derived from different SCs.

Organoid Type	Special Media Components ^#^	Culture Techniques	References
The organoids derived from PSCs			
Intestine	activin A, Wnt-3A, FGF-4, EGF, NOG, R-spondin 1, RA, CHIR99021	Solid matrix; ALI	[42,53,54]
Gastric	activin A, Wnt-3A, FGF4, NOG, gastrin, nicotinamide, EGF, RA, CHIR99021, Y-27632	Solid matrix	[33,55]
Liver	activin A, BMP-4, FGF-4, NOG, FGF-2, HGF, Y-27632, CHIR99021	Solid matrix; ALI	[43,56,57]
Pancreas	activin A, BMP-4, FGF-4, Wnt-3A, NOG, Y-27632, CHIR99021, FGF-10, dorsomorphin, LDN193189, SANT-1, RA, EGF, Nicotinamide, KGF, MSC2530818, ZnSO_4_	Solid matrix; ALI	[43,58]
Lung	Activin A, Wnt-3A, FGF-4, NOG, SB431542, SU5402, SANT-2, SAG, SHH, RA	Solid matrix	[59]
Mammary gland	hydrocortisone, insulin, FGF-10, HGF, pTHrP, FGF-2, heparin, prolactin	Suspension	[60]
Kidney	Activin A, BMP-4, BMP-7, FGF-2, FGF-8, FGF-9, HGF, IDE-1, JAG-1, NOG, Y-27632, CHIR99021, DAPT, IGF-1, Wnt-4, IWR-1, RA, SB431542	Suspension; ALI	[61,62]
Fallopian	activin A, CHIR99021, Y-27632, BMP-4, Wnt-4, Wnt-3A, follistatin, estrogen, progesterone	Solid matrix	[63]
Inner ear	BMP-4, SB431542, FGF-2, LDN193189, Y-27632	Solid matrix	[64]
Thyroid	Activin A, NOG, SB431542, Wnt-3A, KGF, FGF-10, BMP-4, EGF, FGF-2, HSS, IGF-1, insulin, Y-27632, dorsomorphin, CHIR99021, RA, TSH, R-spondin-1	Solid matrix	[30,65]
Blood vessels	Wnt-3A, BMP-4, VEGF-A, CHIR99021, FGF-2, Y-27632, forskolin	Solid matrix	[27]
Optic cup	BMP-4, RA, Wnt-3A, Nodal, DAPT	Suspension	[45]
Brain	Forebrain: dorsomorphine, A83-01, Wnt-3A, CHIR99021, SB431542, BDNF, GDNF, TGF-β, c-AMP. Midbrain: LDN193189, SB431542, SHH, purmorphamine, FGF-8, CHIR99021, BDNF, GDNF, c-AMP. Hypothalamus: LDN193189, SB431542, 1-Thioglycerol, Wnt-3A, SHH, FGF-2 purmorphamine	Suspension	[47,66,67]
The organoids derived from ASCs			
Intestine	EGF, NOG, R-spondin 1, Wnt-3A, JAG-1, Y-27632, CHIR99021, valproic acid	Solid matrix; ALI	[14,41,68]
Gastric	EGF, NOG, R-Spondin 1, Wnt-3A, FGF-10, Y-27632	Solid matrix; ALI	[41,69]
Liver	EGF, NOG, R-Spondin 1, Wnt-3A, FGF-10, HGF, nicotinamide, gastrin	Solid matrix	[70]
Pancreas	EGF, NOG, R-Spondin 1, Wnt-3A, FGF-10, nicotinamide, Y-27632	Solid matrix	[69]
Lung	Wnt3a, R-spondin 1, and NOG, Y-27632, FGF-7, FGF-10, SB202190, KGF, cAMP, monothioglycerol, CHIR99021, ascorbic acid, dexamethasone, IBMX	Solid matrix; ALI	[44,71]
Mammary gland	heparin, EGF, FGF2, insulin, hydrocortisone, cholera toxin, ciproflaxin, Nrg1, NOG, R-spondin 1	Solid matrix	[72,73]
Kidney	EGF, FGF-10, Y-27632, SB431542, A83-01	Solid matrix	[74]
Fallopian	Y-27632, EGF, NOG, FGF-10, nicotinamid, SB431542	Solid matrix	[75]
Bladder	EGF, R-spondin 1, NOG, A83-01, FGF-10, FGF-2, SB202190	Solid matrix; ALI	[25,76]
Prostate	EGF, R-spondin 1, NOG, A83-01, FGF-10, FGF-2, PGE2, SB202190, nicotinamide	Solid matrix	[77]

Notes: # Different combinations of media components may be applied in different labs. SCs: stem cells; PSCs: pluripotent stem cells; ASCs: adult stem cells; FGF: fibroblast growth factor; EGF: epidermal growth factor; NOG: Noggin; RA: retinoic acid; ALI: air-liquid interface; BMP: bone morphogenetic protein; HGF: hepatocyte growth factor; KGF: keratinocyte growth factor; HSS: heparin sodium salt; IGF-1: insulin-like growth factor-1; TSH: thyroid-stimulating hormone; SHH: sonic hedgehog; TGF-β: transforming growth factor-β; SAG: smoothened agonist; pTHrP: parathyroid hormone; JAG-1: Jagged1; VEGF-A: Vascular Endothelial Growth Factor-A; BDNF: brain derived neurotrophic factor; GDNF: glial cell-derived neurotrophic factor; SAG: smoothened agonist, hedgehog agonist; purmorphamine: SHH agonist; SANT-1/2: hedgehog inhibitor; dorsomorphin, A83-01, SB431542, LDN193189: SMAD inhibitors; CHIR99021: GSK-3β inhibitor; Nrg1: neurogulin-1; PGE2: prostaglandin E2.

**Table 3 cells-12-01590-t003:** The functions of main media components applied in organoid cultures.

Special Media Components	Functions
Activin A, BMP-4	As important members of the TGF-β superfamily, Activin A and BMP4 can induce the embryonic stem cells to differentiate into different germ layers.
Wnt-3A	Wnt-3A modulates embryonic development, cell growth, cell differentiation, and tumorigenesis via the canonical Wnt pathway.
Noggin	Noggin regulates the germ-layer-specific derivation of embryonic stem cells and acts as an antagonist of BMP during development.
Y-27632	The ROCK inhibitor improves the survival of human ESC monolayers at the initiation of differentiation.
FGF	The FGF family plays a central role during prenatal development, postnatal growth, and the regeneration of a variety of tissuesby promoting cellular proliferation and differentiation.
EGF	EGF is a protein that stimulates cell growth and differentiation by binding to its receptor.
HGF	HGF has a strong mitogenic ability to regulate cell growth and cell motility on hepatocytes and primary epithelial cells.
Gastrin	Gastrin acts as a growth factor in organoid culture and stimulates the proliferation of cells.
Retinoic acid	Retinoic acid helps to transform cell types from the proliferative profile to the maturation profile by inducing differentiation.
SB202190	A p38 inhibitor induces cardiomyocyte differentiation from human embryonic stem cells.
R-spondin 1	R-spondin proteins are a secreted agonist of the Wnt/β-catenin signaling pathway.
CHIR99021	The small molecule is the GSK 3 inhibitor and the WNT activator, which can promote the differentiation of insulin-producing cells and cardiomyocytes from human PSCs.
Nicotinamide	A water-soluble vitamin is an active component of the coenzymes NAD and NADP and also act as an inhibitor of sirtuins.
SANT-1	SANT-1 is a cell-permeable antagonist that binds directly to smoothened and inhibits the hedgehog signaling to promote β cell differentiation.
LDN193189	A selective BMP signaling inhibitor inhibits the transcriptional activity of the BMP type I receptors ALK2 and ALK3.
Dorsomorphin	An inhibitor of the AMPK and BMP pathways is used to promote special cell differentiation.
KGF	KGF supports ductal specification by upregulating KRT19 and increasing culture homogeneity.
MSC2530818	A WNT inhibitor increases expression.

Notes: TGF-β: transforming growth factor-β; BMP4: bone morphogenetic protein 4; ESCs: embryonic stem cells; FGF: fibroblast growth factor; EGF: epidermal growth factor; HGF: hepatocyte growth factor; ROCK: Rho-associated, coiled-coil containing protein kinase; GSK: glycogen synthase kinase; PSCs: pluripotent stem cells; NAD: nicotinamide adenine dinucleotide; NADP: nicotinamide adenine dinucleotide phosphate; ALK: activin receptor-like kinase; KGF: keratinocyte growth factor; KRT19: keratin 19; CFTCR: cystic fibrosis transmembrane conductance regulator.

## 3. Bone-Related Organoid Culture

As the in vitro model system, organoids are most similar to the in vivo physiology activity in tissues or the body, which promote the development of drug screening, basic research, disease modelling, and precision medicine. Since the establishment of organoid culture technology, organoids such as brain, liver, lung have been established and successfully applied to these fields above. However, the progress of bone organoid culture and establishment is relatively slow because different types of bone cells are located in a special ECM, which is a network of collagen and minerals that keeps changing. Nonetheless, some bone-related organoids have also been investigated and found (Table 4). For the construction of bone organoids, stem cells, the matrix scaffold, and mechanical stimulation are crucial for the growth and differentiation of cellular organoids [51].

### 3.1. Bone-Related Organoids from PSCs

Using human iPSCs and ESCs, cartilaginous organoids were successfully constructed by Tam WL et al. [52]. These cells were first induced into the mesodermal lineage via the addition of CHIR99021, FGF-2 and retinoic acid; subsequently, chondrocytes were differentiated into cartilaginous organoids in a chondrogenic medium consisting of ascorbic acid, β-Mercaptoethanol, TGF-β1, FGF-2, BMP-2 and growth/differentiation factor 5 (GDF5). In addition, murine osteochondral organoids were also constructed by O’Connor SK et al. [87]. They developed an osteochondral organoid using a single murine iPSC in chondrogenic media consisting of TGF-β3, BMP-2, β-mercaptoethanol, ascorbic acid, dexamethasone and β-glycerophosphate, using a dependent scaffold manner. These studies highlighted the promise of PSC technology for cartilage organoids and its potential to be explored for new bone-healing treatment.

Kale, S. [78] cultivated bone spheroids from human osteoblasts that were obtained from bone precursor cells following collagenase treatment, by culturing with serum-free media containing TGF-β1and ITS+. In the study of Buttery LD et al., the differentiation of mouse ESCs toward the osteoblast lineage was found to be stimulated by supplementing different combinations of ascorbic acid, β-glycerophosphat, dexamethasone, retinoic acid or co-culture with fetal murine osteoblasts [79]. In addition, Bielby RC et al. [88] have also shown that human ESCs can be differentiated into osteoblasts with the capacity to form mineralized tissue both in vitro and in vivo, through a culture methodology built for the differentiation of murine ESCs. With the above research, it seems to be a method that is able to form bone spheroids from PSCs. The further development of this method could be a breakthrough in generating sufficient yields of osteogenic cells for use in the understanding of skeletal tissue repair and bone remodeling. In the last year, Li Z A et al. [89] have used different induction media to promote mesenchymal progenitor cell (iMPCs) differentiation, which originated from iPSCs, into different organoids, such as bone, cartilage, adipose, and osteochondral organoids. Further development of this method could be a breakthrough in generating sufficient yields of osteogenic cells for use in the understanding of skeletal tissue repair and bone remodeling.

As the critical component in the bone, bone marrow organoids, whose location contains a lot of hematopoietic cells, stromal cells, and marrow adipose tissue, are also developing [90]. Khan AO et al. [91] created a specific protocol by which to generate and culture bone marrow organoids originating from iPSCs, featuring stroma, lumen-forming sinusoids, and myeloid cells, including proplatelet-forming megakaryocytes. In this particular procedure, iPSCs are aggregated and undergo mesodermal induction with BMP4, FGF2, and VEGFA in the first three days. In the next two days, hematopoietic and vascular lineages are formed under BMP4, FGF2, VEGFA, SCF, and FLT3. Then, cell aggregates are embedded in mixed-matrix hydrogels to support vascular sprouting and are cultured using VEGFA, FGF2, BMP4, SCF, FLT3, IL3, IL6, cGSF, EPO, TPO. After that, vascularized, myelopoietic organoids are produced. In addition, bone marrow organoids also prove that they are an effective tool for modeling blood and bone marrow disorders [92,93].

### 3.2. Bone-Related Organoids from ASCs

Akiva A et al. [94] demonstrated the first complete functional 3D living in vitro model system by which to study early bone formation, both in health and disease, and study formation under the influence of mechanical stimulation or drug administration. First, they isolated bone marrow stromal cells (BMSCs) from one healthy male. Then, these cells were exposed to mechanical stimulation through fluid-flow-derived shear stress, by applying continuous stirring in a spinner flask bioreactor with 3D silk fibroin scaffolds. By employing an osteogenic medium containing ascorbic acid-2-phosphate, dexamethasone, and β-glycerophosphate in this bioreactor, they successfully cultured an early-stage bone (woven bone) organoid. In addition, they noted that the rate of osteocyte differentiation also depended on the glucose concentration in the medium.

### 3.3. Bone-Related Organoids from Bone Precursor Cells

In addition to organoids that are completely differentiated from ASCs, some researchers also have found that the coculture system of adult bone precursor cells and mature cells can form organoids in a special environment. Park Y et al. [80] constructed a trabecular bone organoid by using the coculture system of osteoblasts and osteoclasts in demineralized bone paper (thin sections of demineralized bovine compact bone) activated by Vitamin D3 (VD3) and PGE2. In this system, osteoblasts can acquire the bone lining cell phenotypes and promote bone tissue mineralization. Osteoblasts and osteoclasts are differentiated from osteoprogenitor cells and bone marrow mononuclear cells, respectively. In the processing of differentiation, the osteoblast differentiation medium that includes β-glycerophosphate and ascorbic acid, the osteoclast differentiation medium that contains nuclear factor κB ligand (RANKL), and the macrophage colony-stimulating factor (M-CSF) are used. Furthermore, Iordachescu A et al. [81] also acquired a trabecular bone organoid by seeding primary osteoblasts and osteoclaste from a female into the femoral heads of cattle as the base for trabecular bone organoid generation. The trabecular bone organoid offers a new insight for the study of pathological bone loss, as well as the study of the complex and dynamic regulation of bone remodeling processes.

On the other hand, Torisawa YS et al. [82] established an engineered bone marrow that can keep progenitor cells and hematopoietic stem cells for more than 1 week in culture. To generate a bone-like organoid, they subcutaneously implanted a poly (dimethylsiloxane) (PDMS) device in the back of a mouse. This device has a central cylindrical cavity, which contains type I collagen gels with bone morphogenetic proteins (BMP-2 and BMP-4) and bone-inducing demineralized bone powder (DBP). After 8 weeks, a white cylindrical disk appeared, that is, a bone-like tissue containing a central region of blood-filled marrow. The trabecular bone produced by the engineered bone marrow had a similar natural bone regarding its architectural and compositional properties. Thus, this study offered a method by which to generate define-sized and -shaped bones. It could represent a new method for the study of bone biology, remodeling, and pathophysiology in vitro. Moreover, Serafini M et al. [83] have developed an in vivo model in which the potential of human BMSCs to acquire bone marrow organoids was optimized without exogenous synthetic scaffold. Before transplantation into the subcutaneous tissue of immunocompromised mice, the isolated human BMSCs from healthy children and adults were cultured in chondrogenic differentiation medium with 2-phosphate–ascorbic acid, dexamethasone, and TGF-β1 or TGF-β3. Pievani A et al. [84] used a similar method to establish a bone marrow organoid using human umbilical cord blood-borne fibroblasts from volunteer mothers. Moreover, bone marrow organoids were formed in a high-throughput manner in order to model blood stem cell dynamics [95].

Papadimitropoulos A et al. [85] developed a co-culture system of osteoblasts, osteoclasts, and endothelial cells to mimic the bone turnover process. They isolated endothelial lineage cells and osteoprogenitors from the stromal vascular fraction (SVF) of human adipose tissues, and isolated osteoclast progenitors from human peripheral blood. These cells were cultivated in a perfusion-based bioreactor device with 3D porous ceramic scaffolds and cultured medium, including several typical osteoclastogenic factors, such as M-CSF and RANKL. In a study of developing myeloma in vitro modeling, Visconti RJ et al. [96] also used a coculture system to construct normal bone-like fragments via BMSC-derived osteoblasts and bone marrow macrophage (BMM)-derived osteoclasts.

Nilsson Hall G et al. [86] isolated periosteum-derived cells (PDCs) from the periosteal biopsies of nine donors to form a callus organoid in chondrogenic medium containing ascorbate-2 phosphate, dexamethasone, proline, Y27632, BMP-2, GDF5, TGF-β1, BMP-6, and FGF-2. PDCs not only contain osteoblasts and osteoblastic precursor cells, forming osteocytes and osteoblasts, respectively, but also contain renewable skeletal stem cells that form membranous cortical bone and endochondral bone upon damage [97,98] and possess a higher regenerative ability than BMSCs. The authors showed that multiple callus organoid aggregations resulted in special constructs, forming large bone organs ectopically and healing critical-sized long bone defects in mice. In another recent study, the authors were able to obtain the osteo-callus organoids that originate from BMSCs, with the aid of hydrogel microspheres via digital light-processing (DLP) printing technology. After chondrogenic induction, osteo-callus organoids showed a much higher chondrogenic efficiency and were able to lead to bone regeneration quickly in one month [99].

Gamblin AL et al. [100] used a miniaturized 3D culture with mineral granules to develop osteoblasts and osteoclasts from BMSCs and blood monocytes, respectively. They first seeded BMSCs on particles of biphasic calcium phosphate in osteogenic media supplemented with ascorbic acid, β-glycerophosphate, and dexamethasone. On day 83, osteoclast-pre-differentiated monocytes were added into the 3D constructs of BMSCs, then the 3D co-culture of BMSCs and monocytes was initiated.

### 3.4. Bone-Related Organoids on a Chip

The organ-on-a-chip is a 3D microfluidic cell culture with multiple channels and an integrated circuit (chip), which simulates the activities, mechanics and physiological response of an organ or an organ system, with the aim of replacing animal models [101,102]. In fact, this can be said to be another form of organoid. Apart from brain-on-a-chip, gut-on-a-chip, lung-on-a-chip, etc., bone-on-a-chip also is developing. To study the cell–cell interactions of bone, Mansoorifar A et al. [103] developed a microfluidic platform. The microfluidic device consisted of two or three channels separated by high-resistance posts. Osteoclast precursor cells (RAW264.7) were seeded in one channel, while the other channel had RAW media or RAW media supplemented with RANKL. After undergoing daily fluid shear stress, the stimulated osteocytes promoted osteoclast differentiation. Moreover, Glaser DE et al. [104] presented a bone marrow-on-a-chip model, including perivascular and endosteal niches complete with dynamic, perfusable vascular networks, by using endothelial colony-forming cells and BMSCs. In the review of Zhang Y et al. [105], bone-on-a-chip was well-described regarding the research progress and characterization of this model. It will not be repeated here.

Furthermore, osteoarticular joint-on-a-chip systems have also emerged [106]. For example, to determine the influence of mechanics on cartilage, Paggi C et al. [107] developed a microdevice with the ability to controllably directly induce pressure via actuation chambers on the hydrogel containing chondrocytes. It is of note that Hu Y et al. [108] proposed the concept and schematic diagram of the bone/cartilage organoid on-chip. The authors told us that, as a novel platform of multi-tissue, the bone/cartilage organoid on-chip system has the huge potential to mimic the essential elements, biological functions, and pathophysiological response under real circumstances. In addition, in a review published last year, the authors also reported [109] that the establishment of joint organs-on-chips systems has led to some interesting findings with regard to several joint disorders, such as inflammation and mechanical injuries. However, joint-on-a-chip still lacks the sufficient complexity and physiological relevance required in mechanistic studies.

## 4. Bone Disease Organoid Models

As popular and efficient in vitro models, humanization and physiology are the most significant advantages of organoids. Like other organoids, bone organoids also can better simulate the pathological or physiological environment compared to animal bone disease models, in order to understand the molecule mechanism of disease more precisely. To date, some bone disease organoids have been constructed, and some theories and methods show the huge potential of constructing disease organoid modellings (Figure 3); this organoid culture system could become a promising platform for immunotherapy evaluation and drug screening in precision medicine.

In multiple myeloma (MM), myeloma cells not only suppress bone formation but also promote excessive bone resorption simultaneously. During the process of bone metabolism imbalance, these could appear as osteolytic lesions causing chronic bone pain, cortical bone structural integrity, trabecular reduction, and pathological fractures. Recently, Visconti RJ [96] developed a well-characterized myeloma disease bone organoid in order to study the biological mechanisms of this disease and to search for potential strategies. There are three main steps in this establishment protocol. Firstly, a mineralized endosteal-like ECM was acquired by osteoblasts from BMSCs using Matrigel over 21 days. Next, osteoclasts from a bone marrow macrophage were added to the above cultivated system to improve resorption ability and balance the homeostatic nature for a week. In this step, RANKL and M-CSF were supplied with a completed bone medium to enhance differentiation. Lastly, plasmacytoma cells from human MM were introduced to promote enhanced bone resorption and the formation of osteolytic lesions. After 12 days, the levels of total hydroxyapatite, tartrate-resistant acid phosphatase 5b (TRAcP-5b), type I collagen C-telopeptide (CTX-1), and osteoblastic genes were analyzed. Wei X et al. [110] found that lysine-specific demethylase 1 (LSD1/KDM1A) was the first autosomal dominant MM germline predisposition gene by applying patients’ samples and mouse experiments; they confirmed that KDM1A inhibition increases MM cell line and cell proliferation via MM organoids in a 3D culture.

Osteosarcoma (OS), originating in the osteoblasts, are heterogeneous tumors that are not only associated with chromosomal aberrations and structural abnormalities, but also deletions of entire chromosomal arms or segments. He A et al. [111] developed two protocols (tumor small tissues and single tumor cells) to determine the organoids derived from human primary OS. In the process of determining tissue-derived organoids, the first stage is to mince the isolated tumor into small pieces; then, the minced tissues are washed, resuspended, and layered in a double-layer culture system with a full culture medium, including nicotinamide, A83-01, B-27 without vitamin A, EGF, gastrin, noggin, R-spondin 1, SB-202190 and Wnt-3A. In the process of determining tumor stem cell-originated organoids, single-cell suspensions are generated via filtering with a cell strainer. Next, these cells are embedded in Matrigel and cultured with the same above medium. In addition, the authors also obtained lung metastatic OS organoids using the same methods. However, in the study of Subramaniam D et al., a different method was chosen to construct OS organoids in order to explore the effect of pimozide (a STAT5 inhibitor) [112]. Their organoids were not directly prepared from OS tissues or OS stem cells but were cultured by mixing more than 1000 normal human lung epithelial cells, umbilical vein endothelial cells, normal lung fibroblasts, and normal lymphatic endothelial cells with KHOS/NP GFP-positive OS cells.

The Imbalance between bone mineral resorption and deposition causes osteoporosis. Osteoporosis models can be generated by drugs, surgery, gene knockout or other methods in animals; these processes not only require a long cycle, but also incur a high cost. Cultivating a bone osteoporotic organoid in vitro could greatly save time and finance. The construction of trabecular bone organoids improves the further exploration of trabecular bone niche biology, bone loss and imbalances in fine-level physiological bone remodeling processes [80,81]. In addition, Toni R et al. [113] described in detail the mechanism involved in bone remodeling-related immune cells and the role of endocrine-disrupting chemicals (EDs) as inducers in osteoporosis. The authors summarized these immune cells patterns of response that can recognize presumable sites in early bone lesions. They combine immune cell networks with their bone induction and osteolytic effects after the challenges of basal state and osteoporosis. Indeed, bone resorption is the main characteristic of osteoporosis. EDs may induce either bone deposition inhibition or a bone resorption increase. Collectively, this understanding could assist in determining how different osteoporotic lesions develop and are conditioned, therefore prompting the design of experimental tools for in vitro models, such as those for organoids, in the early stage of the osteoporotic process. For example, researchers could add RANKL/MCSF factors to enhance the activity and number of osteoclasts, which simulate the osteoporosis microenvironment. By micro-CT analysis and TRAP staining analysis, it could be confirmed that osteoporotic organoids are constructed successfully because there are a low mineral deposition and high trap activity in osteoporosis. This could provide a new strategy by which to construct an osteoporosis model.

Many factors such as bacteria, injuries, or bone fractures could cause osteomyelitis. At present, some methods can be used to establish osteomyelitis models, such as the intravenous injection of S. aureus, the injection of S. aureus into the bone marrow cavity, or using implants loaded with S. aureus in animals. These methods are remarkably different from the pathogenicity of clinical factors, and easily lead to the animal’s death due to the over-infection of bacteria [114]. In the research of Yang D et al., it was confirmed that S. aureus can infect human osteocytes, both in infected bone tissue and in cell culture [115]. In addition, there are some 3D in vitro infection models, such as gastric organoid culture with Helicobacter pylori or Escherichia coli, human skin equivalents with S. aureus, and microfluidic 3D models containing osteoblasts within ECM with S. epidermidis [116,117,118,119]. These previous studies offer a great foundation on which to complete the construction of osteomyelitis organoids.

Immune cells, synovial cells, and cytokines are implicated in the pathology of osteoarthritis (OA) and rheumatoid arthritis (RA), but the degrees of the two are not the same. In the physiopathology of OA, changes in the subchondral bone are essential. Kim TW et al. [120] have shown that it is viable to organize chondrogenesis by combining chondrocytes and synovium-derived stem cells (3:1). Maumus M et al. [121] found that inflammatory phenomena are reduced in chondrocytes when synovial cells or mesenchymal cells exist in the culture system. Thus, cytokines such as IL-6, IL-8, or chemokines were reduced in these co-culture models, forming osteochondral spheroids. Moreover, Muraglia A et al. [122] found the osteochondral progenitor spheroids were stimulated for 3 weeks by an osteogenic medium with TGF-β1, dexamethasone, and ascorbic acid. This represents a potential possibility to generate an organoid with a bone and cartilage structure in vitro. In addition, in a study by Broeren et al., it was shown that micromasses were formed with human synovial fibroblast-like synoviocytes or synovial cell suspensions and mononuclear cells in Matrigel. Then, the micromasses were exposed to different inflammatory cytokines that can induce arthritis characteristic phenotypes, such as hyperplasia, fibrosis, and synovial proliferation, in order to acquire different disease modeling. For instance, micromasses were exposed to the pro-inflammatory cytokine tumor necrosis factor alpha (TNF-α) to mimic the synovial membrane in RA. Micromasses were exposed to TGF-β to obtain the synovial membrane in OA. In 2021, Abraham DM et al. [123] successfully produced “mini joint” cultures by taking hybrid skeletal organoids by switching different media at specific time points. They used the skeletal organoid with IL-1β as a model of OA, proving that this or-ganoid could be an effective model with which to mimic bone-related diseases.

It is very important for fractures and bone defects to heal well, especially bone defects, because of the occurrence of amputation in the clinic. Normally, cranium bone or leg bone in animals is artificially destroyed in order to acquire fracture or bone defect models [124]. By constructing bone organoids in vitro, as described by the research of Akiva A et al. [52], scientists can simulate the real bone defect environment without sacrificing animals by destroying the model locally. In addition, callus organoids derived from human PDCs can spontaneously bioassemble into large and engineered tissues in long bone healing after being implanted into mice [86]. In another study, a cartilaginous organoid was shown to promote the healing of critical-sized long bone defects using a scaffold-free method [51].

## 5. Potential Applications of Bone Disease Organoid Models

Organoids are potent in vitro tools used to study tissue biology (e.g., development, homeostasis, regeneration), regenerative medicine and disease modelling (e.g., disease mechanism, drug screening, personalized medicine), and bone organoids also have this research potential in these aspects. In addition, with the increase in the prevalence of cancer and other diseases, and the continuous development of biomedical technology, such as next-generation sequencing, single-cell RNA sequencing, and novel preclinical modeling methods, precision medicine has become an integral part of the national strategic emerging industry. As a novel preclinical modeling method, patient-derived organoids exert an important function in biomarker identification, drug screening, and personalized medicine in patients.

### 5.1. Drug Screening

Before going to market, most drugs treating bone diseases require rigorous in vivo and in vitro testing, such as anti-bone resorption drugs, bone growth-promoting drugs, angiogenic drugs, and anti-inflammatory drugs. In traditional methods, organ toxicity, a long cycle and huge cost are the major drawbacks [125]. Constructing bone organoids will greatly reduce the drug detection cycle and drug toxicity. On the other hand, organoid techniques can enhance cell–cell interactions and can more accurately mimic physiological and pathological conditions have compared to 2D culture. Since patient-derived organoids keep the histological characteristics and heterogeneity of the diseases, organoids represent an ideal model with which to screen new drugs. What is more, organoids exhibit different secretion properties and drug metabolism according to their different environmental cues, providing them with a great foundation on which to enhance drug therapies [126].

Organoid drug screening platforms may be a more practical way to inform patient treatment and serve as a screening tool in clinical trials to accelerate the development of anti-cancer drugs [127]. Many disease organoids have established the biobank for biomarkers, drug testing, drug discovery and so on, such as human gut organoids and breast cancer organoids [128,129], but the biobank concerning bone organoids still needs further development. In an ongoing preclinical trial (NCT03890614), MM markers at different time points of organoid life, the chemosensitivity of patient-derived 3D organoids, and differences in the gene expression and cell markers of the myeloma cells that remained alive after chemotherapy exposure will be evaluated. However, regardless of the advantages of organoid culture, the various techniques used to culture organoids are relatively new and further improvements are needed to enhance drug response and testing [130]. So far, bone organoids provide sufficient resources in phenotypic analysis, or drug testing, especially for the lack of large-scale clinical trials or chronic bone diseases. Therefore, bone organoids are an excellent and ideal model for predicting drug toxicity, measuring chemosensitivity, and screening new drugs.

### 5.2. Precision Medicine

In the clinic, some bone disease patients present non-response symptoms after taking some general medications; this is because several bone diseases (e.g., OS, OA, MM, and RA) are heterogeneous diseases with many different subtypes in terms of their histology and molecule. Thus, it is unsuitable to unify treatment methods for all patients, and precision medicine has increasingly become the focus of research regarding the treatment of bone diseases. Custom and personalized treatment for specific patients expressing an abnormal molecular level is the biggest goal in precision medicine, which aims to maximize efficacy and minimize side effects. The biggest advantage of patient-derived organoids is genetic heterogeneity, in terms of which some cancer organoids have made great progress. Gastric cancer is one such example, with organoids from patients with metastatic cancer of the gastrointestinal tract being cultured and treated with commonly used treatment methods in order to predict the treatment response in the study of Vlachogiannis et al. [131]. They found that organoids have 93% specificity, 100% sensitivity, 88% positive predictive value, and a 100% negative predictive value, which help the clinician to deduce the patient’s response to targeted agents or chemotherapy. Furthermore, they could use patient-derived organoids to mimic tumor-to-patient tumor differences and distinguish the patient’s heterogeneity from chemotherapy drugs. However, the use of bone organoids from patients seems to be a vacancy in personalized medicine. We believe, in the future, that an increasing number of results will prove the advantages of using bone organoids from patients in personalized medicine. Of note is a recent trial concerning novel 3D myeloma organoids obtained from patients with MM in order to study disease biology and chemosensitivity; this will provide personalized medicine and guidance in the setting of relapsed MM (NCT03890614).

## 6. Perspectives and Limitations

Organoids, characterized by a similar physiological activity to that of the body, thus matching the patient-specific disease background, largely improve the success of potential drug candidates and enable faster personalized medicine for patients. The organoid technique partially addresses some issues, such as low-throughput drug screening and high tumor heterogeneity and possesses the promising capability of mimicking physiological or pathological activity and maintaining the patient-specific disease background. Organoids provide a cellular and molecular approach with which to research the contribution of various pathways in the development and homeostasis of different human organs. With the aid of these advantages, a broad variety of organoids have been built for various applications in both translational research and basic studies, but most of these organoid models just represent partial components of a tissue. Controlling the cell type, organization, and cell–cell or cell–matrix interactions is a challenging task in this culture system. Thanks to the flexible design and rapid development of biomaterials and the directional differentiation of stem cells stimulated by special factors, tissue engineering has promising potential applications in organoids. Compared to other types of organoids, the establishment of bone organoids faces more challenges and difficulties in both the organoids themselves and bones. At present, although the development of bone organoids is still in its infancy, scientists are trying various ways to breakthrough these bottlenecks.

The first difficulty is that bone organoids only show a single function for bone, such as bone resorption, bone formation, or hemopoiesis. Although the first complete functional woven bone organoid was constructed, which encouraged researchers to further explore bone organoids, there is still a huge bottleneck regarding the completion of a multifunction analysis in order to satisfy deeper mechanism research in a whole bone organoid. Due to the complexity of various cells’ interaction, it is difficult to control the differentiated direction via the co-culture of several stem cells from different sources. Bone mini organs based on a scaffold-free approach can be seen as building the modeling blocks of 3D printing technology, promising to achieve multifunctional bone tissue. Faced with this challenge, scientists have developed organoid technology and microfluidic organ chips, leading to the vigorous development of the bone organoids field. At the same time, bones are dynamic and subjected to mechanical stimuli. Therefore, by placing bone organoids into perfusion bioreactors or microfluidic chips to mimic in vivo microenvironments, it is possible to summarize the biomechanical relationships between 3D organoids.

Another challenge is vascularization. It is well known that larger tissues and even the body contain rich networks of vascular formations in order to support nutrients and oxygen supplies. For large bone organoids, angiogenesis is necessary. Thus, it is necessary to add extra vascular endothelial cells in the bone organoid culture system to sustain vascularized organoids via the use of bone tissue-engineering material; of note are growth factors such as VEGF and FGF, which should be added to activate the formation of vascular networks.

In addition, calcification and mineralization also are challenges to overcome during bone organoid formation, which occur in native bone. The current bone organoid models do not mimic bone’s nanoscale calcification and its complexity or the cell–matrix and cell–mineral interactions; this is because these models relying on cell-laden hydrogels are not calcified or mineralized. Thrivikraman G et al. [132] developed a biomimetic method to try to address this problem in 2019. They replicated the nanoscale mineralization of 3D bone microenvironments embedded with osteoprogenitor, vascular, and neural cells. The bone microenvironments were cultured in supersaturated calcium and phosphate medium with a non-collagenous protein analog, which guide the deposition of nanoscale apatite. This approach can replicate these definitive characteristics of bone tissue, such as biomineralization, nanostructure, vasculature, and innervation.

Apart from the difficulty of bone organoid formation, determining ways in which to treat bone-related diseases, especially bone-related cancers, is also required. With the advancement of novel genome editing techniques such as CRISPR-Cas9, it is possible that potential therapies for patients might be developed by modifying gene mutation, making epigenetic alterations, genetic variants, or even changes in the chromatin structure of bone cancer-derived organoids. Additionally, due to its versatility, gene-editing tools can help scientists acquire rich models of bone disease for the development of drugs and biomechanical research. In this review, we have discussed individualized or disease-specific bone organoids with candidate gene functions that could be used in some of the original models of histology and carcinogenicity in the future.

In summary, as drug screening tools and therapeutic methods, bone organoids have shown huge potential to mimic development and disease processes.

## Figures and Tables

**Figure 1 cells-12-01590-f001:**
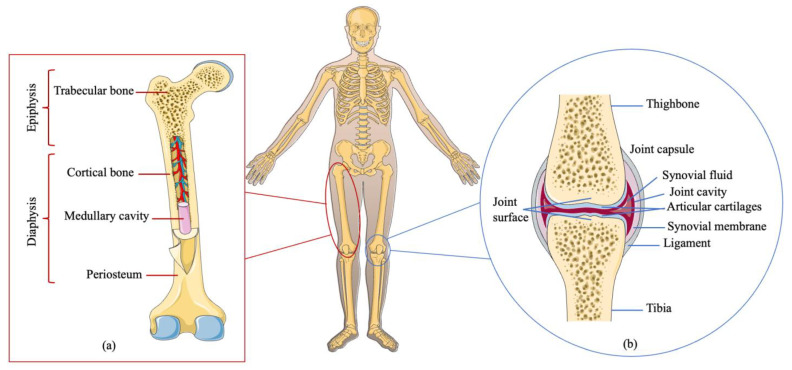
Bone and knee joint anatomy. (**a**) Bone anatomy. Adult bone is composed of cortical bone and trabecular bone. The former, with blood vessels and osteocytes, is a compact bone covering the outer bone surface; the latter, containing bone marrow or fat, constitutes the inner bone. (**b**) Knee joint anatomy. This joint is generally composed of the joint surface, the joint capsule, and the joint cavity.

**Figure 2 cells-12-01590-f002:**
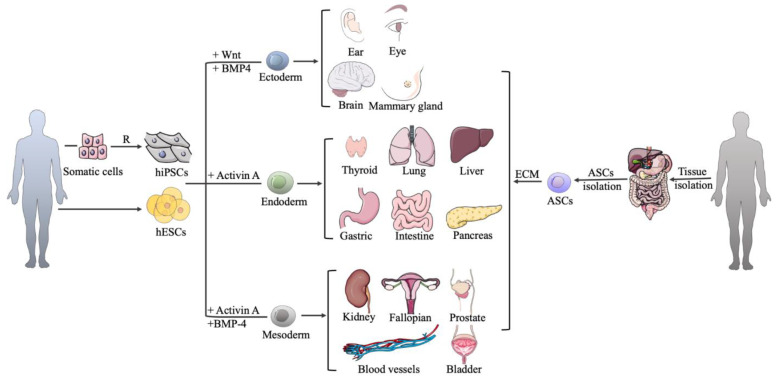
Overview of establishment of human PSC-derived and ASC-derived organoids. PSC-derived organoids first involve three germ-layer specifications (endoderm, mesoderm, or ectoderm) using different factors. ASC-derived organoid cultures require the isolation of the tissue-specific stem cell population, which need to then be embedded into an ECM with defined, tissue-specific combinations of growth factors to allow propagation. R: reprogramming; hiPSCs: human pluripotent stem cells; hESCs: human embryonic stem cells; ASCs: adult stem cells; ECM: extracellular matrix.

**Figure 3 cells-12-01590-f003:**
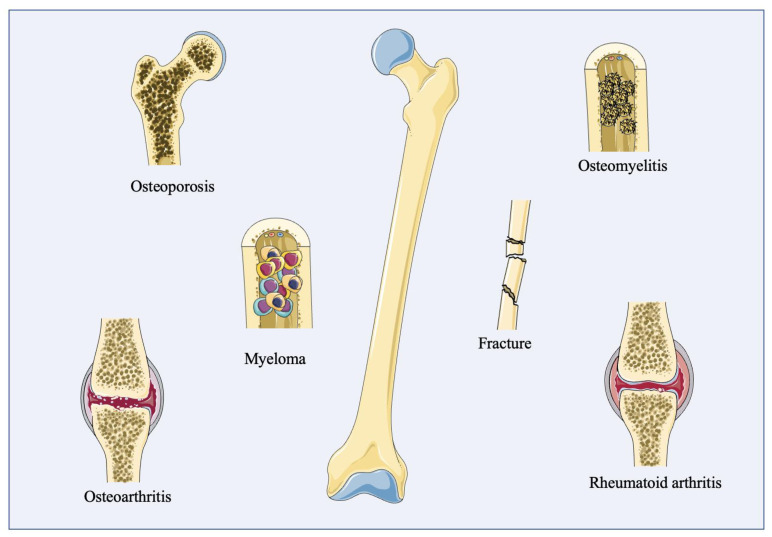
Potential disease modellings in bone organoids.

**Table 1 cells-12-01590-t001:** Summary of bone and joint diseases.

Disease Types	Features	
Bone disorders		
Fracture	a break in the bone caused by stress	
Osteoporosis	a degenerative disease, characterized by the porosity and brittleness of bone in the aged	
Osteomyelitis	an infectious disease caused by the infectious organism *Staphylococcus aureus* (*S. aureus*)	
Osteomalacia	rickets in adults is caused by the inadequate mineralization of the bone	
Osteitis fibrosa cystica	a disease in which bone is replaced by fibrous tissue	
Osteosarcoma	a disorder in which bone tissue grows uncontrollably (either malignant or benign)	
Multiple myeloma	one of the broadest hematologic cancers, characterized by anemia, hypercalcemia, malignant bone infiltration, increased infectious susceptibility, and kidney failure	
Joint disorders		
Osteoarthritis	a ubiquitous noninflammatory degenerative joint disease, characterized by the progressive deterioration of the whole joint	
Bursitis	an inflammation of the lubricating sac located around joints, the synovial bursa, or between tendons and muscles or bones	
Infectious arthritis	a set of arthritis caused by exposure to certain microorganisms	
Rheumatoid arthritis	a chronic, frequently progressive autoinflammatory disease in which inflammation and the thickening of the synovial membranes cause irreversible damage to the joint capsule	
Several other types	psoriatic arthritis	resembles rheumatoid arthritis, but lacks rheumatoid factors in the blood
	ankylosing spondylitis	

**Table 4 cells-12-01590-t004:** Summary of bone-related organoids.

3D Self-Organizing Structures or Organoids	Cells Source	Special Media Components	Culture Techniques	References
Human cartilaginous organoids	Human iPSCs or ESCs	CHIR99021, FGF-2, RA, AA, β-Mercaptoethanol, TGF-β1, FGF-2, BMP-2, GDF5	Solid matrix	[51]
Murine osteochondral organoids	murine iPSC	TGF-β3, BMP-2, AA, β-mercaptoethanol, β-glycerophosphate, dexamethasone	Solid matrix	[78]
Bone spheroids	Osteoblasts	TGF-β1, ITS+	-	[79]
Woven bone organoids	Human BMSCs	AA, dexamethasone, and β-glycerophosphate	Solid matrix	[52]
Trabecular bone organoids	Human osteoblasts and osteoclasts	VD3, PGE2, AA, RANKL, β-glycerophosphate, M-CSF	Solid matrix	[80,81]
Bone marrow organoids	human BMSCs or human CB-BFs	2-phosphate–ascorbic acid, dexamethasone, and TGF-β1 or TGF-β3	in vivo model	[82,83,84]
Bone organoids	Osteoblasts, osteoclasts, endothelial cells	M-CSF, RANKL	Solid matrix	[85]
Callus organoid	PDCs	ascorbate-2 phosphate, dexamethasone, proline, Y27632, BMP-2, GDF5, TGF- β1, BMP-6, FGF-2	Solid matrix	[86]

Notes: iPSCs: induced pluripotent stem cells; ESCs: ASCs: adult stem cells; GDF5: growth/differentiation factor 5; ITS+: containing insulin, human transferrin, and selenous acid; RA: retinoic acid; AA: ascorbic acid; BMSCs: bone marrow stromal cells; RANKL: nuclear factor κB ligand, M-CSF: macrophage colony-stimulating factor, TGF-β: transforming growth factor-β; CB-BFs: umbilical cord blood-borne fibroblasts; PDCs: periosteum-derived cells; FGF: fibroblast growth factor; BMP: bone morphogenetic protein.

## Data Availability

Not applicable.

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
