# Peer review of "Organoids as Innovative Models for Bone and Joint Diseases"

_cells, 2023, doi:10.3390/cells12121590_

Round 1

Reviewer 1 Report

The authors described the organoids and their relevance for the study of bone and joint diseases. Although editing process is required, the review is interesting since it summarizes some studies related to bone-joint organoids model.

The review is well organized; however, the section 1.2 can be deleted and the information inserted in an additional table.

Author Response

Point 1: The authors described the organoids and their relevance for the study of bone and joint diseases. Although editing process is required, the review is interesting since it summarizes some studies related to bone-joint organoids model.

Response 1: Thanks for the overall comments.

Point 2: The review is well organized; however, the section 1.2 can be deleted and the information inserted in an additional table.

Response 2: Thanks for the constructive comments. We have removed this section and retain the information in tabular manner (Table 1).

Reviewer 2 Report

The manuscript’s authors introduce a current and very pertinent theme in literature. The use of organoids is spreading in several areas. However, the area of bone and joint diseases is in a very preliminary stage and still needs bibliographic support.

A general appreciation of the manuscript reveals a lack of care in the presentation, namely, underlining in yellow. What does this mean? In addition, the construction of the sentences has several errors, sentences not starting with capital letters (e.g., in the abstract, bone line 13). Therefore, it is necessary to revise English profoundly.

The authors also revised organoids used in other areas, such as the stomach. Although I find this information pertinent, considering the lack of literature on bone, many bridges are missing for then for bone. What growth factors in other areas might have the potential for the bone area?  What similar methodologies could be performed? Etc. This exercise should be done to make this review article an activity for the area’s future.

In addition, considering the heterogeneity of bone-related diseases, point 4 should be divided by disease typologies. And then, in point 5, in line 502, it talks about a cartilaginous organoid. Shouldn't this part be on line 487, for example, where it talks about "mini joints"? Again, reorganizing the point by types of bone disease would make it easier to read.

Author Response

Point 1: The manuscript’s authors introduce a current and very pertinent theme in literature. The use of organoids is spreading in several areas. However, the area of bone and joint diseases is in a very preliminary stage and still needs bibliographic support. 

 Response 1: Thanks for the overall comments.

Point 2: A general appreciation of the manuscript reveals a lack of care in the presentation, namely, underlining in yellow. What does this mean? In addition, the construction of the sentences has several errors, sentences not starting with capital letters (e.g., in the abstract, bone line 13). Therefore, it is necessary to revise English profoundly. 

Response 2: Thanks for the constructive comments. The yellow part represents the content of the modifications, according to the editor’s suggestion. Marking yellow is to find this content quickly. In addition, we have edited and improved the English language in our revised manuscript.

Point 3: The authors also revised organoids used in other areas, such as the stomach. Although I find this information pertinent, considering the lack of literature on bone, many bridges are missing for then for bone. What growth factors in other areas might have the potential for the bone area?  What similar methodologies could be performed? Etc. This exercise should be done to make this review article an activity for the area’s future.

Response 3: Thanks for the constructive comments. We have added some content to bridge the correlation between other and bone organoids. For example, in lines 274-276, “In bone organoids, growth factors such as CHIR99021, retinoic acid also play the key function in differentiating from iPSCs and ESCs into mesodermal lineage [1].” In lines 309-313, “Although the process of bone organoid formation is similar to that of other organoid formation, the process of bone organoid formation has a higher requirement in many aspects, such as special medium content, matrix, and mechanical stimulation. For example, Akiva A et al [2] employed an osteogenic medium containing ascorbic-acid-2-phosphate, dexamethasone, and β-glycerophosphate to culture woven bone organoid successfully.”

Point 4: Besides, considering the heterogeneity of bone-related diseases, point 4 should be divided by disease typologies. And then, in point 5, in line 502, it talks about a cartilaginous organoid. Shouldn't this part be on line 487, for example, where it talks about "mini joints"? Again, reorganizing the point by types of bone disease would make it easier to read.

 Response 4: Thanks for the constructive comments. We described and summarized bone-disease organoid models by disease classification in point 4. As for point 5, some content has been reorganized and placed in a more suitable location.

1.         Tam, W.L., et al., Human pluripotent stem cell-derived cartilaginous organoids promote scaffold-free healing of critical size long bone defects. Stem Cell Res Ther, 2021. 12(1): p. 513.

2.         Akiva, A., et al., An Organoid for Woven Bone. Advanced Functional Materials, 2021. 31(17): p. 2010524.

Reviewer 3 Report

This manuscript is a comprehensive review on bone and joint organoids for modeling osteoarticular diseases in vitro. Although the English is poorly written, this topic is of high interest for readers of the journal. The following issues should addressed by the authors prior to publication: 

1) Abstract (l. 11-13): The authors wrote 'Bone and joint diseases are the fourth...' but only mentioned cardiovascular and cancers. They have probably omitted diabetes. The sentence has too many wording repetitions. Please correct. 

2) Introduction (l. 31-32): The authors stated that 'the inner trabecular bone ... devotes to the movement of joints'. This is incorrect. It should be related to hematopoeisis within the bone marrow niche.  Please correct accordingly. 

3) Line 59: Rephrase the sentence as following ' The imbalance between osteoblasts/osteoclasts in bone remodeling or fibroblast/macrophage synoviocytes in articular joints induces bone/joint-related diseases such as osteoporosis or osteoarthritis.'  

4) section 1.2 Bone and joint diseases (l. 74): Bone fracture heals in the majority of cases while cartilage cannot repair. The cascade events in fracture healing should be introduced here. 

5) line 77 and along the manuscript: Please write S. aureus, in vitro and in vivo in italic.  

6) Section 1.3 Comparison of research models in vitro (l. 112-116): Please rewrite these sentences in correct English. 

7) Section 2.2 Organoid culture techniques (l. 181-184): Please rewrite this sentence in correct English. 

8) Section 3 Bone-related organoid culture: The literature search is not complete and the authors should cite the following reports that attempted or reviewed 3D culture of bone.

Gamblin AL, Renaud A, Charrier C, Hulin P, Louarn G, Heymann D, Trichet V, Layrolle P. Osteoblastic and osteoclastic differentiation of human mesenchymal stem cells and monocytes in a miniaturized three-dimensional culture with mineral granules. Acta Biomater. 2014 Dec;10(12):5139-5147. 

Marturano-Kruik A, Nava MM, Yeager K, Chramiec A, Hao L, Robinson S, Guo E, Raimondi MT, Vunjak-Novakovic G. Human bone perivascular niche-on-a-chip for studying metastatic colonization. Proc Natl Acad Sci U S A. 2018 Feb 6;115(6):1256-1261. 

Mansoorifar A, Gordon R, Bergan R, Bertassoni LE. Bone-on-a-chip: microfluidic technologies and microphysiologic models of bone tissue. Adv Funct Mater. 2021 Feb 3;31(6):2006796. 

Glaser DE, Curtis MB, Sariano PA, Rollins ZA, Shergill BS, Anand A, Deely AM, Shirure VS, Anderson L, Lowen JM, Ng NR, Weilbaecher K, Link DC, George SC. Organ-on-a-chip model of vascularized human bone marrow niches. Biomaterials. 2022 Jan;280:121245. 

Zhang Y, Yu T, Ding J, Li Z. Bone-on-a-chip platforms and integrated biosensors: Towards advanced in vitro bone models with real-time biosensing. Biosens Bioelectron. 2022 Oct 13;219:114798. 

9) The authors should add a section on osteoarticular joint on a chip and cite the following articles:

Jorgensen C, Simon M. In Vitro Human Joint Models Combining Advanced 3D Cell Culture and Cutting-Edge 3D Bioprinting Technologies. Cells. 2021 Mar 8;10(3):596. 

Li ZA, Sant S, Cho SK, Goodman SB, Bunnell BA, Tuan RS, Gold MS, Lin H. Synovial joint-on-a-chip for modeling arthritis: progress, pitfalls, and potential. Trends Biotechnol. 2023 Apr;41(4):511-527. 

Hu Y, Zhang H, Wang S, Cao L, Zhou F, Jing Y, Su J. Bone/cartilage organoid on-chip: Construction strategy and application. Bioact Mater. 2023 Jan 20;25:29-41. 

10) Section 6 Perspectives and limitations: The authors should consider other limitations of current bone organoids towards future directions. In particular, the integration of calcified or mineralized extracellular matrices is often overlooked. The current methods that rely on cell-laden hydrogels are either not mineralized or simply combined with calcium and phosphate micro/nanoparticles. The current bone organoid models do not mimic the coordinated and rapidly moving mineralization front found in bone tissue. Other important limitations are vascularization and innervation of bone models. A recent article attempted to address this issue but is not cited: 

Thrivikraman G, et al. , Rapid fabrication of vascularized and innervated cell-laden bone models with biomimetic intrafibrillar collagen mineralization. Nature Communications, 2019. 10(1): p. 3520.

Author Response

This manuscript is a comprehensive review on bone and joint organoids for modeling osteoarticular diseases in vitro. Although the English is poorly written, this topic is of high interest to readers of the journal. The following issues should addressed by the authors prior to publication: 

1) Abstract (l. 11-13): The authors wrote 'Bone and joint diseases are the fourth...' but only mentioned cardiovascular and cancers. They have probably omitted diabetes. The sentence has too many wording repetitions. Please correct. 

Response 1: Thanks for the constructive comments. We have modified this sentence. In lines 11-14, “Bone and joint disease are the fourth most widespread disease in addition to cardiovascular disease, cancer, and diabetes, which seriously affect people’s quality of life. Bone organoid seems to be a great model to improve the treatment of bone and joint disease in the future.”

2) Introduction (l. 31-32): The authors stated that 'the inner trabecular bone ... devotes to the movement of joints'. This is incorrect. It should be related to hematopoeisis within the bone marrow niche.  Please correct accordingly. 

Response 2: Thanks for the constructive comments. We have modified this sentence. In lines 31-32, “whereas the inner trabecular bone is active in metabolism and related to hematopoiesis within the bone marrow niche.”

3) Line 59: Rephrase the sentence as following ' The imbalance between osteoblasts/osteoclasts in bone remodeling or fibroblast/macrophage synoviocytes in articular joints induces bone/joint-related diseases such as osteoporosis or osteoarthritis.'  

Response 3: Thanks for the constructive comments. We have modified this sentence. In lines 64-66.

4) section 1.2 Bone and joint diseases (l. 74): Bone fracture heals in the majority of cases while cartilage cannot repair. The cascade events in fracture healing should be introduced here. 

Response 4: Thanks for the constructive comments. We have removed this section (1.2) and retained the information in a tabular manner, according to reviewer 1 comments.

5) line 77 and along the manuscript: Please write S. aureus, in vitro and in vivo in italic.  

Response 5: Thanks for the constructive comments. We have removed this section (1.2) and retained the information in a tabular manner, according to reviewer 1 comments.

6) Section 1.3 Comparison of research models in vitro (l. 112-116): Please rewrite these sentences in correct English. 

Response 6: Thanks for the constructive comments. We have modified these sentences. In lines 102-106, “However, it is challenging to mimic bone in vitro because the bone always stays in dynamic equilibrium by osteoclasts and osteoblasts balance. Simple models such as 2D cultures are great models for studying cell functions in vitro, but it is difficult to truly simulate cell functions and signaling pathways in vivo without the interactions of cell-cell and that of cell-matrix making.”

7) Section 2.2 Organoid culture techniques (l. 181-184): Please rewrite this sentence in correct English. 

Response 7: Thanks for the constructive comments. We have modified these sentences. In lines 242-245, “Surprisingly, synthetic hydrogels solve these problems, which have been added into organoid culture (cerebral and intestinal) to replace natural matrices [3, 4], but these hydrogels need to custom-tailor to meet specific requirements in different organoids owing to the low bioactive.”

8) Section 3 Bone-related organoid culture: The literature search is not complete and the authors should cite the following reports that attempted or reviewed 3D culture of bone.

Gamblin AL, Renaud A, Charrier C, Hulin P, Louarn G, Heymann D, Trichet V, Layrolle P. Osteoblastic and osteoclastic differentiation of human mesenchymal stem cells and monocytes in a miniaturized three-dimensional culture with mineral granules. Acta Biomater. 2014 Dec;10(12):5139-5147. 

Marturano-Kruik A, Nava MM, Yeager K, Chramiec A, Hao L, Robinson S, Guo E, Raimondi MT, Vunjak-Novakovic G. Human bone perivascular niche-on-a-chip for studying metastatic colonization. Proc Natl Acad Sci U S A. 2018 Feb 6;115(6):1256-1261. 

Mansoorifar A, Gordon R, Bergan R, Bertassoni LE. Bone-on-a-chip: microfluidic technologies and microphysiologic models of bone tissue. Adv Funct Mater. 2021 Feb 3;31(6):2006796. 

Glaser DE, Curtis MB, Sariano PA, Rollins ZA, Shergill BS, Anand A, Deely AM, Shirure VS, Anderson L, Lowen JM, Ng NR, Weilbaecher K, Link DC, George SC. Organ-on-a-chip model of vascularized human bone marrow niches. Biomaterials. 2022 Jan;280:121245. 

Zhang Y, Yu T, Ding J, Li Z. Bone-on-a-chip platforms and integrated biosensors: Towards advanced in vitro bone models with real-time biosensing. Biosens Bioelectron. 2022 Oct 13;219:114798. 

Response 8: Thanks for the constructive comments. We have added these references. In lines 579-593, “The organ-on-a-chip is a 3-D microfluidic cell culture with multi-channel and integrated circuit (chip), which simulates the activities, mechanics and physiological response of an organ or an organ system [5, 6]. In fact, this can be said to be another form of organoids. Apart from brain-on-a-chip, gut-on-a-chip, lung-on-a-chip, etc., bone-on-a-chip also is developing. To study the communications of bone cell-cell, Mansoorifar A  et al [7] developed a microfluidic platform. The microfluidic device consisted of 2 or 3 channels separated by high-resistance posts. osteoclast precursor cells (RAW264.7 ) were seeded in one channel while the other channel had RAW media or RAW media supplemented with RANKL. After undergoing daily fluid shear stress, the stimulated osteocytes promoted osteoclast differentiation.

Moreover, Glaser DE et al [8] presented a bone marrow-on-a-chip model including perivascular and endosteal niches complete with dynamic, perfusable vascular networks, by using endothelial colony-forming cells and BMSCs. In the review of Zhang Y et al [9], bone-on-a-chip was so well-described the research progress and characterization of this model. It will not be repeated here.”

9) The authors should add a section on osteoarticular joint on a chip and cite the following articles:

Jorgensen C, Simon M. In Vitro Human Joint Models Combining Advanced 3D Cell Culture and Cutting-Edge 3D Bioprinting Technologies. Cells. 2021 Mar 8;10(3):596. 

Li ZA, Sant S, Cho SK, Goodman SB, Bunnell BA, Tuan RS, Gold MS, Lin H. Synovial joint-on-a-chip for modeling arthritis: progress, pitfalls, and potential. Trends Biotechnol. 2023 Apr;41(4):511-527. 

Hu Y, Zhang H, Wang S, Cao L, Zhou F, Jing Y, Su J. Bone/cartilage organoid on-chip: Construction strategy and application. Bioact Mater. 2023 Jan 20;25:29-41. 

Response 9: Thanks for the constructive comments. We have added these references. In lines 594-605, “Furthermore, osteoarticular joint-on-a-chip systems have also emerged [10]. For example, to know whether mechanical influence on cartilage, Paggi C et al [11] developed a microdevice with the ability to controllably induce pressure by actuation chambers directly on the hydrogel containing chondrocytes.  Note of, Hu Y et al [12] proposed the concept and schematic diagram of the bone/cartilage organoid on-chip. The authors told us as a novel platform of multi-tissue, the bone/cartilage organoid on-chip system has the huge potential to mimic the essential elements, biological functions, and pathophysiological response under real circumstances. Besides, in a review published last year, the authors also said [13] the establishment of joint organs-on-chips systems has led to some interesting findings on several joint disorders, such as inflammation and mechanical injuries. However, joint-on-a-chip still lacks the sufficient complexity and physiological relevance required in mechanistic studies.”

10) Section 6 Perspectives and limitations: The authors should consider other limitations of current bone organoids towards future directions. In particular, the integration of calcified or mineralized extracellular matrices is often overlooked. The current methods that rely on cell-laden hydrogels are either not mineralized or simply combined with calcium and phosphate micro/nanoparticles. The current bone organoid models do not mimic the coordinated and rapidly moving mineralization front found in bone tissue. Other important limitations are vascularization and innervation of bone models. A recent article attempted to address this issue but is not cited: 

Thrivikraman G, et al. , Rapid fabrication of vascularized and innervated cell-laden bone models with biomimetic intrafibrillar collagen mineralization. Nature Communications, 2019. 10(1): p. 3520.

Response 10: Thanks for the constructive comments. We have modified section 6, in lines 901-940, and added the reference, in lines 925-931. “The first difficulty is that bone organoids only show a single function for bone, such as bone resorption, bone formation, or hemopoiesis. Although the first complete functional woven bone organoid was constructed, which encouraged researchers to further explore bone organoids, it is still a huge bottleneck to complete multifunction to satisfy deeper mechanism research in a whole bone organoid. Due to the complexity of various cells interaction, it is difficult to control the differentiated direction via the co-culture of several stem cells from different sources. Bone mini organs based on a scaffold-free approach can be seen as building modeling blocks of 3D printing technology, promising to achieve multifunctional bone tissue. Faced with the challenge, scientists developed organoid technology and microfluidic organ chips, leading to the vigorous development of the bone organoid field. At the same time, bones are dynamic and subject to mechanical stimuli. Therefore, by placing bone organoids into perfusion bioreactors or microfluidic chips to mimic in vivo microenvironment, it is possible to summarize the biomechanical relationships between 3D organoids.

Another challenge is vascularization. It is well known that larger-sized tissues and even the body contain rich networks of vascular formation to support nutrients and oxygen supplies. For large bone organoids, angiogenesis is necessary. Thus, it is necessary to add extra vascular endothelial cells in the bone organoid culture system to sustain vascularized organoids by using bone tissue-engineering material, note that growth factors such as VEGF and FGF should be added to activate the formation of vascular networks.

In addition, calcification and mineralization also are challenges to overcome during bone organoid formation, which occur in native bone. The current bone organoid models do not mimic bone’s nanoscale calcification and its complexity, and the cell–matrix, and cell–mineral interactions, because these models rely on cell-laden hydrogels that are not calcified or mineralized. Thrivikraman G et al [14] developed a biomimetic method to try to address this problem in 2019. They replicated the nanoscale mineralization of 3D bone microenvironments embedded with osteoprogenitor, vascular, and neural cells. The bone microenvironments were cultured in supersaturated calcium and phosphate medium with a non-collagenous protein analog, which guides the deposition of nanoscale apatite. This approach can replicate these definitive characteristics of bone tissue such as biomineralization, nanostructure, vasculature, and innervation.

Apart from the difficulty of bone organoid formation, how to treat bone-related diseases, especially bone-related cancers, also needs to be overcome. With the advancement of novel genome editing techniques like CRISPR-Cas9, it is possible to be a potential therapy for patients by modifying gene mutation, epigenetic alterations, genetic variants, or even changes in the chromatin structure of bone cancer-derived organoids. Additionally, due to its versatility, gene-editing tools can help scientists acquire rich models of bone disease for the development of drugs and biomechanical research."

Round 2

Reviewer 2 Report

The authors answered the questions and improved the article accordingly.